# Hypertension induced by pregnancy and neonatal outcome: Results from a retrospective cohort study in preterm under 34 weeks

**Marta David Rocha de Moura**[1]\*, **Paulo Roberto Margotto**[1], **Karina Nascimento Costa**[2], **Maria Rita Carvalho Garbi Novaes**[3]

1 Neonatal Intensive Care Unit, Brasília Mother and Child Hospital, Asa Sul, Brasília, Brazil, 2 Department of Child and Adolescent Medicine, University of Brasília, UnB, Brasília, Brazil, 3 Higher School of Health Sciences, SMHN Conjunto A B 01 FEPECS—Asa Norte, Brasília, Brazil

\* marta.rocha@escs.edu.br

## Abstract

### Objective

The present study seeks to assess the impact of gestational hypertensive disorders on premature newborns below 34 weeks and to establish the main morbidities and mortality in the neonatal period and at 18 months.

### Materials and methods

A retrospective observational study was carried out with 695 premature newborns of gestational age (GA) between 24 and 33 weeks and 6 days, born alive in the Neonatal ICU of Brasília's Mother and Child Hospital (HMIB), in the period from January 1, 2014, to July 31, 2019. In total, 308 infants were born to hypertensive mothers (G1) and 387 to normotensive mothers (G2). Twin pregnancies and diabetic patients with severe malformations were excluded. Outcomes during hospitalization and outcomes of interest were evaluated: respiratory distress syndrome (RDS), brain ultrasonography, diagnosis of bronchopulmonary dysplasia (BPD), diagnosis of necrotizing enterocolitis, retinopathy of prematurity, breast-feeding rate at discharge, survival at discharge and at 18 months of chronological age and relationship between weight and gestational age.

### Results

Newborns with hypertensive mothers had significantly lower measurements of birth weight and head circumference. The G1 group had a higher risk small for gestational age (OR 2.4; CI 95% 1.6–3.6; p <0.00), as well as a greater risk of being born with a weight less than 850 g (OR 2.4; 95% CI 1.2–3.5; p <0.00). Newborns of mothers with hypertension presented more necrotizing enterocolitis (OR 2.0; CI 95% 1.1–3.7); however, resuscitation in the delivery room and the need to use surfactant did not differ between groups, nor did the length of stay on mechanical ventilation, or dependence on oxygen at 36 weeks of gestational age.

**Data Availability Statement:** All the data are available for inquiry or analysis. In https://doi.org/10.6084/m9.figshare.15078798.v1.

**Funding:** The authors received no specific funding for this work.

**Competing interests:** The authors have declared that no competing interests exist.

Survival was better in newborns of normotensive mothers, and this was a protective factor against death (OR 0.7; 95% CI 0.5–0.9; p <0.01). In the follow-up clinic, survival at 18 months of chronological age was similar between groups, with rates of 95.3% and 92.1% among hypertensive and normotensive mothers, respectively. Exclusive breastfeeding at discharge was 73.4% in the group of hypertensive women and 77.3% in the group of normotensive mothers. There were no significant differences between groups.

## Conclusion

Among the analyzed outcomes, arterial hypertension during pregnancy can increase the risk of low weight, small babies for gestational age (SGA), deaths in the neonatal period and enterocolitis, with no differences in weight and survival at 18 months of chronological age. Arterial hypertension presents a high risk of prematurity in the neonatal period, with no difference at 18 months of age.

## Introduction

Hypertensive Disorders of Pregnancy (HDP) present a serious complication that affects approximately 2.5 to 3.0 percent of women, increasing the risk of maternal and neonatal complications [1, 2]. Worldwide, hypertensive disorders remain the leading cause of maternal mortality related to pregnancy [1].

Hypertensive disorders of pregnancy appear as a hypertensive condition that develops at any time after 20 weeks of pregnancy, accompanied or not by proteinuria. Among the manifestations of these syndromes is eclampsia, which presents with a convulsive component, and HELPP syndrome, which manifests with the presence of hemolysis, elevated liver enzymes and thrombocytopenia; HELLP is a severe form of preeclampsia and not a separate disorder [3, 4].

Hypertensive Disorders of Pregnancy cannot be prevented; therefore, the identification of maternal risk factors becomes an important obstetric mission [1]. The risk factors associated with the development of gestational hypertension, previous history of pre-eclampsia, primiparity, obesity, family history of pre-eclampsia, number of previous pregnancies (if any) and chronic medical conditions, such as hypertension and diabetes, maternal age greater than or equal to 40 years, obesity, diabetes mellitus, chronic kidney disease, systemic lupus erythematosus, presence of antiphospholipid, multiple pregnancies and high altitude. In addition, it is important to highlight the action of transplant of live kidneys, which can increase the risk of developing pre-eclampsia in the recipients by up to six-fold more than in non-transplanted women [1–6].

The decision to induce labor must balance maternal and neonatal risks; thus, the basic objective of obstetrics is to carefully prolong pregnancy to improve the perinatal outcome without compromising maternal safety [7, 8]. Some authors suggest that expectant control of pregnancy is associated with an increase in maternal complications in a pregnancy of 32 weeks or more [7–9].

The neonatal complications described here range from prematurity to fetal growth restriction. The latter is the most frequent neonatal complication in newborns (NB) with hypertensive mothers [10, 12–14]. Perinatal mortality rates in growth-restricted neonates are 6 to 10 times that of those with normal growth [2, 5]. Doppler ultrasonography in fetuses of hypertensive mothers is a way of assessing the severity of intrauterine growth restriction and intrauterine monitoring allows the disease progression to be observed non-invasively [11].

Other morbidities described in newborns of mothers with hypertensive conditions are bronchopulmonary dysplasia (BPD), retinopathy of prematurity (ROP), sepsis and longer time on mechanical ventilation. Hematological changes such as thrombocytopenia and leukopenia are also frequently described in the literature [10].

Exposure to HDP may be associated with an increased risk of autism spectrum disorder (ASD) and attention deficit/hyperactivity disorder [12, 13]. These findings highlight the need for greater pediatric surveillance of newborns exposed to HDP to allow early interventions that can improve neurological development outcomes.

The aim of this study is to assess the impact of gestational hypertensive disorders on premature newborns below 34 weeks and to identify the mortality rate and morbidities at discharge and at 18 months of corrected gestational age.

## Material and methods

### Study design, setting and population

This retrospective cohort study included preterm infants, with gestational age (GA) between 24 and 33 weeks and 6 days, born alive in a maternity center in Brasília, Federal District, Brazil, at the Mother and Child Hospital (Hospital Materno Infantil de Brasília, HMIB), and admitted to the Neonatal Intensive Care Unit of this Hospital, from January 1, 2014, to July 31, 2019.

HMIB is a Teaching and Referral Hospital in Federal District. The hospital This hospital was opened in1964 and it is a referral center since 1988 for neonates, women, and surgical children as tertiary hospital. Having a medical residency program in the areas of pediatrics, neonatology, pediatric intensive care, obstetrics, fetal medicine and pediatric surgery.

A total of 18768 deliveries in the institution during the specified period. Preterm born in the period was 1765 newborns (Fig 1).

Exclusion criteria were premature infants transferred directly from the delivery room to another service, as well as deaths found in the delivery room, twins, newborns from other services and patients with severe malformations, diabetics mothers and NB that have not undergone umbilical artery doppler flowmetry. All patients included in the study were followed up during hospitalization and evaluated for the outcomes of interest at 28 days of life or at discharge/death/transfer and at 18 months of corrected gestational age.

### Sample size calculations and statistical analysis

In The study design followed the STROBE checklist recommendations of the Equator Network [15]. Sample size calculations and statistical analysis. Sample size calculations were done in Epi Info TM version 7.2.2.6 (Centers for Disease Control and Prevention (CDC); USA) at 80% power and 95% confidence interval. Based on a 10% prevalence of hypertension during pregnancy, considering a 99.9% confidence level, 90% power and a 1:1 ratio (exposed and unexposed). The estimated sample size was 432 newborns, including 216 newborns with hypertensive mothers and 216 newborns with normotensive mothers. During the study period, it was possible to select 695 NBs for the study, divided into two groups, G1—hypertensive pregnant women, with 308 newborns, and G2—normotensive pregnant women with 387 newborns.

### Operational definitions

In this study, hypertensive disorders of pregnancy were classified as preeclampsia, gestational hypertension, chronic hypertension and preeclampsia superimposed on chronic hypertension.

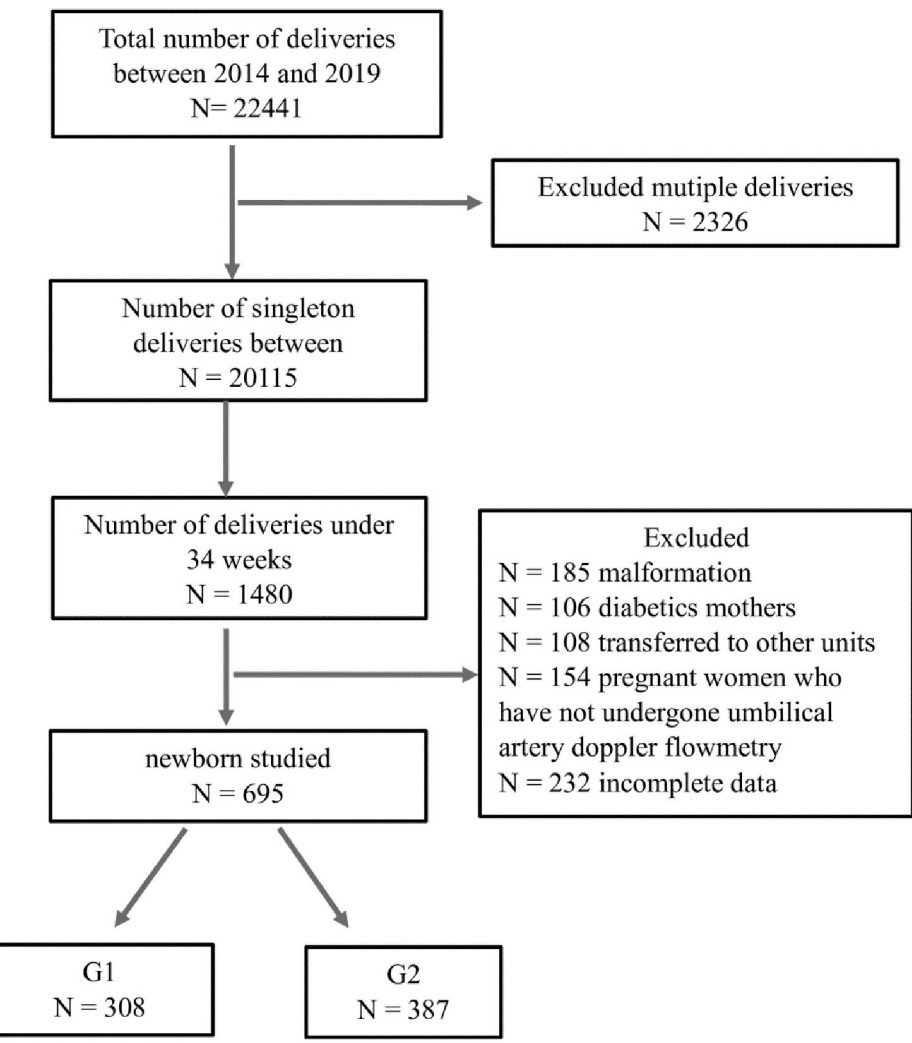

**Fig 1. Schematic diagram showing the number of participants medical birth registry, 2014–2019.**

Hypertension in pregnancy was defined as systolic blood pressure (BP) ≥140 mmHg and/ or a diastolic BP ≥ 90 mmHg. Preeclampsia was defined as characterized by a BP of 140/90 mm Hg or greater after 20 weeks' gestation in a woman with previously normal BP and who have proteinuria. Eclampsia was defined as seizures that cannot be attributable to other causes, in a woman with preeclampsia and chronic hypertension is that which is present before pregnancy or diagnosed before 20 weeks of pregnancy [3].

Assess the fetal outcomes of hypertensive disorders of pregnancy, data was obtained from the medical records of the study subjects included the following gestational data, obtained from obstetric notes and confirmed in the postpartum period: prenatal care (> 2 consultations for premature births); maternal age; maternal infection (clinical chorioamnionitis or peripartum urinary tract infection); antenatal steroid use (> one dose administered before delivery, regardless of the type of steroid); use of magnesium sulfate, diagnosis of hypertensive disease of pregnancy (including variations of pre-eclampsia, eclampsia, HELLP syndrome and pre-eclampsia overlapping with chronic arterial hypertension); and absent end-diastolic velocity waveform in gestational ultrasound up to 72 hours before and during delivery.

The data related to birth were gestational age defined by the best obstetric estimate, preferably the precise date of the last menstruation, followed by early obstetric ultrasound (up to the 18th week of pregnancy). In the absence of such data, gestational age (GA) was estimated by examining the newborn, performed immediately after birth or within 12 hours, using the method of New Ballard [15]. Other data analyzed were gender; birth weight; weight and gestational age classification to define small for gestational age (SGA), according to the criteria of Lubchenco et al. [16] and Margotto [17], from 29 weeks; Apgar of the 1st and 5th minutes; need for resuscitation in the delivery room defined using positive pressure ventilation under a mask; and the use of surfactant in the delivery room.

Regarding neonatal, the variables analyzed were: respiratory distress syndrome (RDS) with clinical and radiological diagnosis; ventilatory assistance time (continuous positive airway pressure—nasal CPAP and/or mechanical ventilation); early ($<72$ h life) or late ($> 72$ h) sepsis; brain alterations diagnosed by brain ultrasonography; diagnosis of bronchopulmonary dysplasia (BPD) defined by the need for supplemental oxygen at 36 weeks of postmenstrual age; death or hospital discharge. Diagnosis of necrotizing enterocolitis was defined by the presence of at least one clinical sign: bilious gastric aspirate or vomiting; abdominal distension; either occult or evident blood in the stool without anal fissure and the presence of at least one radiological sign included intestinal pneumatosis, gas in the hepatobiliary system and/or pneumoperitoneum.

Brain ultrasonography was performed on all newborns who survived up to 7 days. In newborns who had more than one examination, the worst ultrasound finding was considered. Cystic leukomalacia was defined by the presence of small cavities close to the lateral ventricles, and peri-intraventricular hemorrhage was classified according to the criteria of Papile et al. [18] and considered severe when it reached grade III or IV. The examinations were performed with GE® equipment by two neonatologists who specialized in diagnostic imaging, responsible for the routine assessment of preterm infants younger than 34 weeks of gestation.

Data analysis after completeness was checked, data were coded and entered Microsoft Excel and exported to statistical software SPSS version 22 for analysis. We used descriptive statistics to describe parameters collected from the files. Bivariate and multiple logistic regression were done to determine factors associated with unfavorable perinatal outcome. Variables which did not show statistical significance in the bivariate analysis were excluded from the multivariate analysis. P-value less than 0.05 and 95% confidence interval not including 1 were considered statically significant.

## Ethical consideration

Ethical approval was by the Research Ethics Committee by the Foundation and Teaching and Research in Health Sciences—FEPECS, of the Health Department of the Federal District under registration CAAE 58280716.9.0000.5553, approval number 2,137,741. Regarding informed consent, the ethics committee waived the requirement for informed consent; however, confidentiality was maintained.

## Results

In the period from January 1, 2014, to July 31, 2019, 22,441 babies with gestational age of less than 37 weeks were born in the Federal District, with 1480 newborns admitted to the Neonatal ICU of the HMIB during this period. Of these, 695 met the inclusion criteria for the study. A total of 785 newborns (NB) were excluded for the following reasons: multiple malformations (85); gestational diabetes (206); transferred (108) and pregnant women with inconclusive data (154) and incomplete data (232). Of the 695 newborns included in the study, 308 were born to hypertensive mothers (G1), and 387 were born to normotensive mothers (G2).

In G1 diagnosis of hypertensive disease of pregnancy was 49,0% (151/308 pregnant women), pre-eclampsia 16,6% (51/308 pregnant women), eclampsia 15,6% (48/308 pregnant women), HELLP syndrome 7,1% (22/308 pregnant women) and pre-eclampsia overlapping with chronic arterial hypertension 10,1% (31/308 pregnant women) and chronic hypertension in 1,6% (5/308 pregnant women).

Table 1 describes the clinical characteristics presented by the women studied in G1 and G2. There was no difference between groups regarding gestational age and the use of antenatal corticosteroids. The use of magnesium sulfate was frequent in G1, and cesarean delivery was more common in this group.

Table 2 shows the morbidity of the newborns studied. The anthropometric measurements of birth weight and head circumference were significantly lower in G1 and had a higher risk of SGA NB (OR 2.4; CI 95% 1.6–3.6; p <0.00), as well as a greater risk of being born with a weight less than 1000 g (OR 3.9; 95% CI 2.8–5.4; p <0.00). Newborns in group G1 had more necrotizing enterocolitis (OR 2.0; CI 95% 1.1–3.7; p<0.03), but resuscitation in the delivery room and the need to use surfactant and oxygen dependence at 36 weeks of gestational age did not differ between groups. However, the G1 stay more time on mechanical ventilation (Table 2).

The changes in brain ultrasound until the first week of life, as well as the worst ultrasound image described before discharge or death, did not differ between groups (Table 3).

Being born in group G2 was a protective factor against death during the hospitalization period (OR 0.7; 95% CI 0.5–0.9; p <0.01), however, no difference was observed at 18 months of chronological age, with rates of 95.3% in G1 and 92.1% in G2. Exclusive breastfeeding at discharge was 73.6% in the group of hypertensive women and 77.7% in the group of normotensive mothers. There were no significant differences between groups (Table 3).

## Discussion

In a classic study, Villar et al. [19] compared the perinatal results in the following subgroups: pre-eclampsia and SGA, gestational hypertension and SGA, and SGA not explained as a reference group. Women with pre-eclampsia and SGA had a higher risk of their newborns staying in the neonatal ICU and experiencing neonatal death after adjusting the study for the location and socioeconomic status, but this difference disappeared after adjusting for birth weight and gestational age. This suggests that the excess risk associated with pre-eclampsia and SGA can be explained by these variables [19].

**Table 1. Characteristics of the studied pregnant women divided considering the presence or not of Hypertensive disorders of pregnancy.**

| | G1 | G2 | p | OR (95% CI) |
|---|---|---|---|---|
| | n: 308 NB | n: 387 NB | | |
| Maternal Age (years) | 28.2 ± 6.8 | 26.1 ± 6.9 | 0.00 | - |
| Number of prenatal consultations | 5.3 ± 2.4 | 5. ± 2.4 | 0.63 | - |
| Gestational Age | 28.9± 2.9 | 29.8 ± 2.2 | 0.00 | - |
| Cesarean delivery% | 68.5 | 34.4 | 0.00 | 0.2 (0.2 to 0.3) |
| Urinary infection % | 12.0 | 23.0 | 0.00 | 0.4 (0.3 to 0.7) |
| Chorioamnionitis% | 13.0 | 25.3 | 0.00 | 0.4 (0.3 to 0.6) |
| Prenatal corticosteroids% | 77.3 | 73.1 | 0.21 | 1.2 (0.8 to 1.8) |
| Magnesium Sulfate% | 59.7 | 36.4 | 0.00 | 2.6 (1.9 to 3.5) |

Data: mean ± SD, p-Value < 0,05 via Student't–test; Data: n (percent),

p–Value < 0,05 via $X^2$ test

**Table 2. Neonates' morbidity considering the presence or not of Hypertensive disorders of pregnancy.**

| | G1 | G2 | p | OR |
|---|---|---|---|---|
| | n: 308 NB | n: 387 NB | | (95% CI) |
| Birth weight (grams) | 1054.2± 377.3 | 1321.8 ± 352.5 | 0.00 | - |
| Head circumference (cm) | 27.8 ± 4.2 | 28.8 ± 3.2 | 0.00 | - |
| Mechanical ventilation (days) | 7.3 ± 15.6 | 5.2 ± 13.0 | 0.05 | - |
| Apgar 1 min | 5.9 ± 2.1 | 6.3 ± 2.1 | 0.02 | - |
| Apgar 5 min | 7.8 ± 1.6 | 7.8 ± 1.5 | 0.97 | - |
| Hospitalization time in days | 37.8 ± 32.3 | 36.7 ± 39.5 | 0.76 | - |
| Weight at hospital discharge | 1919.5 ± 226.9 | 1.811.9 ± 346.1 | 0.00 | - |
| NB <850 g % | 23.7 | 11.4 | 0.00 | 2.4 (1.4 to 3.5) |
| SGA rating% | 23.7 | 10.1 | 0.00 | 2.4 (1.6 to 3.6) |
| Male % | 50.3 | 48.8 | 0.70 | 1.0 (0.8 to 1.4) |
| Neonatal resuscitation % | 66.6 | 61.5 | 0.17 | 1.2 (0.9 to 1.7) |
| Use of surfactant% | 47.7 | 41.6 | 0.10 | 1.3 (0.9 to 1.7) |
| Early sepsis% | 34,1 | 39.3 | 0.17 | 0.8 (0.6 to 1.1) |
| Later sepsis% | 37.7 | 31.0 | 0.07 | 1.3 (1.0 to 1.8) |
| Use of O2 at 36 weeks% | 23.0 | 24.4 | 0.81 | 0.9 (0.6 to 1.5) |
| Necrotizing Enterocolitis% | 8.4 | 4.4 | 0.03 | 2.0 (1.1 to 3.7) |
| Retinopathy of prematurity% | 13.6 | 15.0 | 0,63 | 0.8 (0.5 to 1.4) |
| Leukomalacia (%) | 6.3 | 2.1 | 0.03 | 3.1 (1.1 to 8.8) |

HIV: Intraventricular Hemorrhage, Data: n (percent), p–Value< 0,05 via $X^2$ test

Data: mean ± SD, p-Value < 0,05 via Student't–test; NB–New Born; SGA—Small for gestational age.

It was observed that most women attended prenatal care; however, normotensive mothers had fewer consultations (5.3 ± 2.4 and 5.0 ± 2.4, p = 0.63). Adequate monitoring of the pregnant woman is extremely important in preventing both maternal and fetal morbidity and mortality. Webster et al. [20], in a systematic review with meta-analysis, suggest that antihypertensive treatment reduces the risk of severe hypertension in pregnant women with chronic hypertension and, consequently, fewer maternal and neonatal complications. However, the lack of randomized controlled trials does not clarify which is the best therapeutic option [20].

Hypertensive pregnant women were older, with a mean age of 28.2 ± 6.8 compared to normotensive women, 26.1 ± 6.9 (p = 0.00). The prevalence of gestational hypertensive disease increases proportionally with the increase in maternal age and occurs more commonly in women over 34 years old; the data collected by this study demonstrate the same trend [21, 22].

**Table 3. Survival at discharge and at 18 months in neonates considering the presence or not of hypertensive disorders of pregnancy.**

| | G1 | G2 | p | OR (IC 95%) | |
|---|---|---|---|---|---|
| | n: 308 RN | n: 387 RN | | | |
| Survival at discharge (%) | 69.8 | 77.3 | 0.03 | 0.7 | (0.5 to 0.9) |
| Exclusive breastfeeding at discharge (%) | 73.4 | 77.7 | 0.25 | 0.8 | (0.5 to 1.1) |
| Survival at 18 months (%) | 95.3 | 92.1 | 0.15 | 1.7 | (0.9 to 3.7) |
| Breastfeeding at 18 months (%) | 66.7 | 70.9 | 0.28 | 0.8 | (0.6 to 1.1) |
| Weight at 18 months (kg) | 8.868 ± 0.720 | 8.830 ± 0.670 | 0.72 | | |

Data: mean ± SD, p-Value < 0,05 via Student't–test / Data: n (percent), p–Value< 0,05 via $X^2$ test.

The use of antenatal corticosteroids above 70% in both groups reflects obstetric care; the benefits of this therapy are described in the literature in aspects such as decreased risk of neonatal death, the occurrence of respiratory distress syndrome and intraventricular hemorrhage [21, 22].

Magnesium sulfate has been well evaluated as an effective intervention in the treatment of women with preeclampsia, the drug of choice for the peripartum care of women with preeclampsia and eclampsia [23–25]. It was observed that in the studied group, 59.7% of hypertensive pregnant women used magnesium sulfate, but only 36.4% of normotensive pregnant women. The data presented show that obstetric team needs to be made aware of the need to use magnesium sulfate in all pregnant women at risk of premature birth [23–25].

Cesarean delivery was the most used mode of delivery in hypertensive pregnant women 68.5% of cases (OR 0.2; 95% CI 0.2–0.3). However, the American College of Obstetrics and Gynecology (ACOG) recommends vaginal delivery for this group of pregnant women [26]. It is suggested that vaginal delivery is safer than operative delivery in women with pre-eclampsia and should be attempted. The isolated diagnosis of pre-eclampsia should not be seen as an immediate indication for cesarean delivery; however, hypertensive pregnancy disease may be accompanied by other problems such as fetal distress and anomalous presentations, which constitute an indication for cesarean section [26].

Chorioaminionitis were more frequent in G2, presenting 25.3%, and 13.0% in G1 (p <0.00). We can consider that sudden infectious conditions led to premature birth in newborns (NB) in the control group of this study. Although the causes of approximately half the cases of prematurity are of unknown etiology, infectious conditions are classically described as risk factors for prematurity [27].

The urinary infection was also more frequent in G2 23.0% and only in 12.0% G1. Gagliardi L et al. [28] define the etiological background of very/extremely preterm birth can be divided into two main categories: intrauterine infection/inflammation and placental vascular dysfunction. The first category is associated with chorioamnionitis (CA), preterm labor, premature rupture of membranes (PROM), placental abruption, and cervical insufficiency, whereas the second category is associated with gestational hypertensive disorders and condition known as fetal indication/fetal growth restriction [28].

Studies have pointed to the high risk of adverse outcomes in pregnant women who have overlapping pathologies [3–5]. It was not possible in this study to consider the subtypes diagnosed in the different groups under a more detailed analysis, as the choice was made not to stratify the samples for an individualized analysis of the cases.

However, it was observed that the newborns born to hypertensive mothers had lower birth weight (1054.2 ± 377.3 grams) compared to controls (1321.8 ± 352.5 grams p = 0.00). The same occurred with head circumference (27.8 ±4.2 cm in G1 and 28.8 cm ± 3.2 in G2, p <0.00). In addition, infants born to hypertensive mothers had a higher frequency of intrauterine growth restriction, with 23.7% being classified as small for gestational age (SGA), compared to the control group, at 10.1% (OR 2.4; 95% CI 1.6–3.6). These findings are in accordance with studies that found the same association between maternal arterial hypertension and NB with growth restriction [5, 12, 28].

The weight at discharge, showed an asymmetrical distribution. In this context, the median observed in G1 1970 grams and G2 1980 grams with p = 0.75; however, there was no significant difference between both groups.

When evaluating the babies again, this time at 18 months of age, an attempt was made to identify whether gestational hypertensive disease would be a risk factor for growth disorders in low-weight preterm infants. For this, logistic regression models were constructed, controlled by gestational age and sex, including maternal hypertension. In the classification between

birth weight and gestational age, it was observed that HDP was not a risk factor for inadequate weight increase (OR = 0.54; 95% CI: 0.21–2.1) at 18 months of corrected age. Similar data were observed by Kiy et al. [29], in a cohort of premature infants at 18 months.

It was identified in the present study that gestational arterial hypertension increased the chance of SGA by approximately two and a half times (OR = 2.4; 95% CI: 1.6–3.6).

Considering this was a retrospective study, there was a difficulty in finding adequate records of height. The ordering of the main results of this study alerts us to the higher frequency of growth disorder in low-birth-weight preterm children born to hypertensive mothers [8, 10, 11].

There were no differences regarding the need for resuscitation and the diagnosis of respiratory distress syndrome, or the need for exogenous surfactant. Although a significantly increased risk for RDS was not found, some authors suggest that this type of association is possible [10, 11].

Maternal hypertension and prematurity can lead to immaturity in the fetal gastrointestinal tract, poor vascular supply, and alteration of the intestinal microbiota, which in turn can be associated with a cascade of events, culminating in the development of necrotizing enterocolitis (NEC) [29–31]. In the present study, a significant association was observed between hypertensive maternal disease and neonatal NEC OR of 3.8 (95% CI 1.7–8.6).

There were no differences between the groups regarding the duration of use of mechanical ventilation, nasal CPAP and oxygen at 28 days or 36 weeks of gestational age, demonstrating that respiratory changes are not linked to maternal pressure changes, but to the complications and limitations of prematurity and low birth weight.

The literature is conflicting about the effects of pre-eclampsia and the development of retinopathy of prematurity (ROP), one of the main causes of childhood blindness worldwide. No differences in the diagnosis of ROP between groups were observed in this study [32]. Retinopathy of prematurity is initially manifested with delayed physiological vascular development of the retina, followed by aberrant vasoproliferation, and is highly correlated with extreme prematurity and low postnatal growth [33, 34].

In the studied sample, there were no differences between groups regarding the presence of cerebral hemorrhage, however we observed that leukomalacia was more frequent in G1 (OR 3.1 CI 95% 1.1–8.8). We also observed that the use of magnesium sulfate was low even in the group of hypertensive pregnant women 58,7% in G1. This situation can be explained by the characteristics of the HMIB, which receives pregnant women from other maternity hospitals and many already arrive outside the intervention condition.

The presence of cerebral hemorrhage can be identified by brain ultrasound and are associated with an increased risk of neurodevelopmental disorders, which may inform the need for greater pediatric surveillance of babies exposed to maternal hypertensive disease. Ensuring early stimulation can help improve the outcome of neurological development; therefore, the neurological prognosis for these premature babies whose mothers have high blood pressure is still very controversial [10, 12].

Although the diagnosis of early neonatal sepsis is more frequent in G2, no statistical difference was observed between groups (OR 0.8 of CI 95% 0.6–1.1); the same was found in the diagnosis of late sepsis, which was more frequent in G1, but without statistical difference OR 1.3 of CI 95% 1.0–1.8.

Higher neonatal mortality was observed in the group of hypertensive mothers, confirming the results of previous studies [8, 10, 11, 35, 36]. When the results were adjusted for birth weight, the presence of hypertension was not shown to be an isolated risk factor for death (OR 1.3 of 95% CI 0.8–1.9). This result follows what has been described in the literature.

There are important limitations of this study that must be considered when interpreting the reported findings. As the project was of a retrospective nature, suggesting that other

confounding factors may be operating and have not been identified. However, it was still possible to detect significant and clinically relevant differences between preterm infants born to hypertensive mothers compared to normotensive mothers.

The results of this study show that gestational hypertensive disorder has short-term repercussions in premature newborns, who are born more frequently by cesarean section and are at greater risk of being born small for gestational age and of presenting necrotizing enterocolitis. Studies with a larger sample size of a multicentric and prospective nature with late follow-up are necessary to consolidate these findings.

In these NB studies, it is possible to see that the population of newborns with Hypertensive Disorders of Pregnancy treated at the HMIB had more mortality and morbidity. We recommend careful monitoring of hypertensive pregnant women with a multidisciplinary and careful approach, guiding the follow-up of these women and their children to improve the clinical results of these newborns.

## Acknowledgments

We gratefully appreciate the support of medical doctors in the Neonatal Intensive Care Unit of the Brasília Mother and Child Hospital.

## Author Contributions

**Conceptualization:** Marta David Rocha de Moura.

**Data curation:** Marta David Rocha de Moura.

**Formal analysis:** Marta David Rocha de Moura.

**Investigation:** Marta David Rocha de Moura.

**Methodology:** Marta David Rocha de Moura.

**Project administration:** Maria Rita Carvalho Garbi Novaes.

**Resources:** Karina Nascimento Costa.

**Supervision:** Paulo Roberto Margotto, Karina Nascimento Costa, Maria Rita Carvalho Garbi Novaes.

**Validation:** Paulo Roberto Margotto, Karina Nascimento Costa.

**Writing – original draft:** Marta David Rocha de Moura.

**Writing – review & editing:** Paulo Roberto Margotto, Karina Nascimento Costa, Maria Rita Carvalho Garbi Novaes.

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
