## [Decision Letter · Decision Letter 0]

7 Dec 2020

PONE-D-20-32227

HYPERTENSION INDUCED BY PREGNANCY AND NEONATAL OUTCOME: A RETROSPECTIVE COHORT STUDY

PLOS ONE

Dear Dr. ROCHA,

Thank you for submitting your manuscript to PLOS ONE. After careful consideration, we feel that it has merit but does not fully meet PLOS ONE’s publication criteria as it currently stands. Therefore, we invite you to submit a revised version of the manuscript that addresses the points raised during the review process.

We look forward to receiving your revised manuscript.

Kind regards,

Antonio Simone Laganà, M.D., Ph.D.

Academic Editor

PLOS ONE

Journal Requirements:

2. In the ethics statement in the manuscript and in the online submission form, please provide additional information about the patient records used in your retrospective study, including: a) whether all data were fully anonymized before you accessed them; or b) whether the ethics committee waived the need for informed consent. If patients provided informed written consent to have data from their medical records used in research, please include this information.

3. or more information on PLOS ONE's expectations for statistical reporting, please see https://journals.plos.org/plosone/s/submission-guidelines.#loc-statistical-reporting. Please update your Methods and Results sections accordingly.

4.We note that you have indicated that data from this study are available upon request. PLOS only allows data to be available upon request if there are legal or ethical restrictions on sharing data publicly. For information on unacceptable data access restrictions, please see http://journals.plos.org/plosone/s/data-availability#loc-unacceptable-data-access-restrictions.

5. Please amend your list of authors on the manuscript to ensure that each author is linked to an affiliation. Authors’ affiliations should reflect the institution where the work was done (if authors moved subsequently, you can also list the new affiliation stating “current affiliation:….” as necessary).

6. Please ensure that you refer to Figure 1 in your text as, if accepted, production will need this reference to link the reader to the figure.

7. Please include a copy of Table xxxx which you refer to in your text on page 11.

Additional Editor Comments (if provided):

The topic of the manuscript is interesting. Nevertheless, the reviewers raised several concerns: considering this point, I invite authors to perform the required major revisions.

Reviewers' comments:

Reviewer's Responses to Questions

**Comments to the Author**

1. Is the manuscript technically sound, and do the data support the conclusions?

Reviewer #1: Yes

Reviewer #2: No

Reviewer #3: Partly

Reviewer #4: Partly

2. Has the statistical analysis been performed appropriately and rigorously? 

Reviewer #1: Yes

Reviewer #2: No

Reviewer #3: No

Reviewer #4: No

3. Have the authors made all data underlying the findings in their manuscript fully available?

Reviewer #1: Yes

Reviewer #2: Yes

Reviewer #3: No

Reviewer #4: No

4. Is the manuscript presented in an intelligible fashion and written in standard English?

Reviewer #1: No

Reviewer #2: No

Reviewer #3: No

Reviewer #4: No

5. Review Comments to the Author

Reviewer #1: I applaud the effort. You've amassed a fairly large cohort. There are many things however that need to be addressed. Some are described here, more in the attached PDF with comments/edits.

The writing is full of new abbreviations, some of which are not introduced, some are redundant or conflicting, and some are just unnecessary confusion. You use multiple different abbreviations to describe the 2 groups, when G1 & G2 would suffice. I've changed some, noted others, but this is very confusing to the reader and must be cleaned up. Other abbreviations appear without explanation (e.g., RN), or are explained & not used. You also interchange GHS & SHG?

I'm very confused about your exclusion/inclusion criteria. You exclude a large number of babies because of lack of Doppler (Table), but in the results you indicate that only ~ 70% of subjects had Doppler studies. I cannot understand this dichotomy; either it is an exclusion criteria, or it isn't.

There are substantial problems with your footnotes. I suspect during revisions you may have scrambled them up. I tried to correct a couple, but you need to go thru the paper and assure that the footnote numbers in it actually are to the appropriate references.

It appears that diastolic flow reversal is a critical contributor to the poor outcomes. I think you need to expand on & emphasize this. Perhaps even break G1 into DR/DZ vs normal Doppler patterns. Perhaps DR/DZ is the critical factor, not the maternal blood pressure. Did any G2 mother's have DR/DZ? Did this affect their fetuses?

Reviewer #2: I was pleased to revise the manuscript entitled “HYPERTENSION INDUCED BY PREGNANCY AND NEONATAL OUTCOME: A RETROSPECTIVE COHORT STUDY” (Manuscript Number: PONE-D-20-32227).

Study approval was viewed by the institutional review board.

I was particularly pleased to review this paper. In my honest opinion, the topic is interesting enough to attract the readers’ attention. Nevertheless, authors should clarify various points and improve the manuscript. Moreover, they should better discuss limitations of the study that are not evidenced in the discussion.

In general, the Manuscript may benefit from extensive revisions, as suggested below:

- All the text needs a language revision by a native English speaker person, in order to improve its readability, typos, and grammatical errors.

- I would suggest revising the manuscript, starting from the abstract, using more appropriate terms and definitions. Hypertensive pregnancy syndrome is not a term usually used to define the spectrum of hypertensive disorders of pregnancy. Low weight should be used as Low birth weight, small for gestational age, and low birth weight are not the same definition. Only one is reported in results.

- I would suggest checking the use of abbreviations.

- Introduction is not appropriate. It does not provide a clear overview of available pieces of evidence on the topic and does not clarify the gap in the literature that the current study aims to cover. Overall, it does not help to clarify the study aim, which answer is sought and why.

- The sample size calculation needs reporting the study hypothesis used to calculate the sample size. In other words, clarify what the calculated sample size was estimated.

- As a retrospective observational study, I would suggest providing details regarding the source of the list of patients, the source of data, how and who extracted data of patients.

- The “independent” variables are actually the dependent variables. The independent variable is the exposure or not exposure to the hypertensive disorder of pregnancy.

- Discussion should start referring to the study aim and providing an overall view of all results. Here it is reported only a secondary result.

- Overall, introduction and study aim, study methods, study results, and discussion does not appear consistent. The manuscript is hard to follow and, in some points, confusing. It cannot be accepted in the present form.

Reviewer #3: Issue 1: Firstly, with the introduction, clearly state what void this study is filling in the literature. In the discussion, try to avoid listing results over again. Instead focus on putting the research and most salient findings into the context of preexisting literature. I would focus on the outcomes that relate back to the AIM/hypothesis or illuminate an interesting contradiction with current published data. For example, the impact of chorioamnionitis on non-hypertensive preterm group should be emphasized as a distinct etiology of preterm birth, whereas it is less of the impetus for delivery in the hypertensive group. If available in your data set, please comment on the incidence of eclampsia within your population.

While the ability to have doppler data in a study that also has significant neonatal granularity is a strength of the study, I think it needs clarification. The inclusion of doppler information and the reasoning should be better stated in the introduction. In the discussion, this causal relationship should be clearer: are abnormal dopplers an effect modifier? What is the effect of abnormal dopplers in the setting without hypertensive disease? This needs to be clearly defined. Please be specific and use commonly accepted terminology for doppler findings (example: Umbilical artery dopplers may be normal, elevated, absent, or reversed diastolic flow).

Additionally, in the discussion the literature for administration of BMZ and magnesium is reviewed extensively. This could be summarized more succinctly as it is a well-accepted interventions and were not a main focus of the design and outcomes of your study. The authors should highlight the uniqueness of some of the study design: for example, having data on neonates 18 months out is rare. Even though maternal hypertension did not end up being a risk factor for poor growth at 18 months, this is clinically interesting as it implies a “catch up of growth” phenomenon (ie although born small they join an improved growth curve once delivered). Please expand on this.

Issue 2: Secondly, the methods section could benefit from additional clarification: how was chart abstraction done? Were people trained to ensure accurate collection? Were charts audited to check internal validity? The protocol used at their institution for deciding who gets umbilical artery dopplers and management should be clarified. Since small for gestational age was a major finding of their study, they should expand on their explanation of the criteria used (Are Lubchenco et al and Margotto population or customized curves? Is their population similar to those in this study?). For the statistical analysis, consideration should be made in adjusting for confounders like gestational age when looking at hospitalization, birth weight etc.

Issue 3: I would consider decreasing the strength of the language used to report the conclusions. Specifically, it was reported that with maternal hypertension, there was a decreased risk of survival. However, this association disappeared/was no longer significant when corrected for birth weight (CI crosses 1).

Issue 4: Lastly, the language is unclear and difficult to follow. In general, please utilize abbreviations accepted by international professional groups, such as ACOG. Please see some suggestions below for revisions. It may be helpful to work with an editor. Please use accepted nomenclature and abbreviations throughout your paper

Grammar/language:

- Results:

o When presenting data of odds ratios include the 95% CI and p value

o Paragraph 2: Zero diastole� change to absent diastole

o Paragraph 5: doesn’t really show morbidity and mortality or neonates

- Figures:

o Titles for table 1 and 2- GH and GNH are confusing, consider renaming to abbreviations consistent with for what they stand in English

o Consider simplifying numbers of words for table 1 and 2 titles

o Consider adding Doppler characteristics in table 1

o Table 1:

P values could go out past two decimal points or rounded

Consider adjusting the weights for GA

o Table 2:

Consider putting your units in the title of the columns if they are all in precents

Clarification of “resuscitation at sunrise” do you mean birth?

Define your abbreviations (SGI, NB)

o Table 3:

Please re-naming your title as this is more than just survival outcomes

Clarify the meaning of “survival to high”

It may improve the readability to have it in the format: OR (95% CI, p=…).

Column IC 95% should be 95% CI

- Introduction:

o Second paragraph, last sentence: Consider rewording to say “White coat hypertension is an additional category…”

o Third paragraph: please add “accompanied by lab abnormalities indicative of end organ damage or by proteinuria” (reference ACOG Executive Summary 2013)

o Fourth paragraph: consider changing to “risk factors associated with development of pregnancy induced hypertension include: previous history of pre-eclampsia ….

o Paragraphs 6-8 could be combined into one as the subject is maternal and neonatal complications with hypertensive disease of pregnancy.

o Ninth paragraph: define SHG

- Materials and methods:

o First paragraph: could simplify this sentence and remove passive language

o Could move your exclusion criteria to the second paragraph to keep your inclusion/exclusion criteria together

- Discussion:

o Mode of delivery: Think about refining the way your frame this issue-I wouldn’t say it is “preferred” but the risk of cesarean section is increased due to worsening maternal disease or fetal distress

o Paragraph 12: 3rd sentence is difficult to understand. Where is the data from intrauterine growth restriction coming? Based on the birth weight, you can say they are SGA, but not IUGR

o Paragraph 8: You state “the low use of magnesium sulfate among normotensive women in this study is worrying”. This needs clarification as there are multiple indications for magnesium (tocolysis, neuroprotection and seizure prophylaxis). For example, many of the hypertensive patients likely received magnesium for seizure prophylaxis in the setting of severe pre-eclampsia. Whereas, in a normotensive group, the patients are more likely to receive it for other indications.

Reviewer #4: The authors report a meaningful analysis of a population that warrants further study, the neonatal outcomes associated with pregnancy induced hypertension in mothers of premature infants < 34 weeks. However, the translation into English is often awkward and thus needs significant editing. It will be confusing to use the Spanish abbreviations if the article is published in English, especially when the abbreviations are not used consistently. (GH, GHA, GN, GNH).

The title of the manuscript should reflect that the population being reported are premature infants < 34 weeks born to mothers with PIH. There are certainly many pregnant women with PIH who deliver from 34 weeks onward, and the outcomes for their infants may be very different from the population described.

In the Abstract Results and the Results sections, the CI95% for NEC is written incorrectly (should be 1.7 – 8.6)

The introduction section is too long. The discussion of terminating pregnancy seems unnecessary as this is rarely an option for the gestational ages of the study population.

Apgar scores (not “Apgars”) are categorical, not continuous and not normally distributed variables and need to be analyzed as such using non-parametric analysis.

Definition of pregnancy induced hypertension is not clear.

Not sure why mothers without Doppler evaluations were excluded.

Not sure why over 100 babies were referred elsewhere. If they belong predominantly to one study group (hypertensive or normotensive) this should be reported.

Not sure why authors used death defined at 28 days as opposed to survival to hospital discharge.

Don’t understand how mortality could be affected by PIH, but not survival at 18 months. It seems that the 18 months statistic is survival from NICU discharge to 18 months of age.

Unclear whether diagnosis of necrotizing enterocolitis requires at least one clinical sign and one radiological sign.

The phrase “head ultrasound” or “cranial ultrasound” should be used instead of “skull ultrasound”

There should be a more detailed description of the logistic regression analyses used. More results of the effect of umbilical flow should be presented in the Results section, or in another Table, but should not appear only in the Discussion section.

The Discussion section is far too long and overreaching in its scope. It is suggested that the authors focus on the findings of their study that might present new insights into outcomes of PIH in this population of premature infants, findings that are novel or else those that corroborate other studies that might impact the care of pregnant women with PIH and the outcomes of their infants when born prematurely.

In Table 3, don’t understand the meaning of “survival to high%”

6. PLOS authors have the option to publish the peer review history of their article (what does this mean?). If published, this will include your full peer review and any attached files.

Reviewer #1: No

Reviewer #2: No

Reviewer #3: No

Reviewer #4: No

---

## [Author Response · Author response to Decision Letter 0]

15 Apr 2021

Dear Reviewers, 

 Through this, we saw the submission of the manuscript, corrected and revised entitled: “Hypertension induced by pregnancy and neonatal outcome: Results from a retrospective cohort study in preterm under 34 weeks.” under the authorship of , Marta David Rocha de Moura, Paulo Roberto Margotto, Karina Nascimento and Maria Rita CG Novaes for your appreciation.

rewritten article suggestions and very pertinent comments, thus contributing to improve the quality of our article. After a detailed analysis of the comments and questions, as well as the errors pointed out and suggestions contained in the opinions sent to us, the article has undergone some changes, which are indicated below.

1. Is the manuscript technically sound, and do the data support the conclusions?

The summary and the introduction have been changed in order to accommodate the request of the reviewers. The justification, which appears in the last paragraph of the introduction, prior to the objective, was reformulated as follows to meet the reviewers' request. 

2. Has the statistical analysis been performed appropriately and rigorously?

A revision was made in the Material and Methods in order to add such information.

In the first paragraph of the Material and Methods, the following excerpts were included, in order to meet the reviewer's request. 

3. Have the authors made all data underlying the findings in their manuscript fully available?

I understand the importance of the requested authors. made the study database available at Dryad is an open source, community driven project that takes a unique approach to data publication and digital preservation. Being found in the register ROCHA, MARTA; Carvalho Garbi Novaes, Maria Rita; Margotto, Paulo Roberto; Nascimento Costa, Karina (2021), HYPERTENSION INDUCED BY PREGNANCY AND NEONATAL OUTCOME: A RETROSPECTIVE COHORT STUDY, Dryad, Dataset, https://doi.org/10.5061/dryad.47d7wm3cn

4. Is the manuscript presented in an intelligible fashion and written in standard English?

In addition to the corrections, the text underwent a profound grammatical revision and correction, made by an American professor of English and a biologist appointed by the University of Brasilia, in addition to having previously been sent to the American Journal Experts (AJE) where word adjustments were made. and spacing.

5. Review Comments to the Author

Reviewer #1. 

Thank you for all the recommendations presented. Adjustments were made to the body of the text to meet requests. An orthographic review and adjustment in the text and abbreviations presented in the article were made. It was really unclear in the text initially sent and some terms were translated and confused in the English language. We clarify that umbilical artery dopllerfluxometry was only performed in G1 - newborns of hypertensive mothers. Exclusion criteria were better described in an attempt to clarify blind spots.

Reviewer #2: 

Thank you very much for the excellent placements, the spelling and grammar was adjusted by an American proofreader appointed by the University of Brasilia. The abbreviations were standardized, there were really serious structural errors that left the term confused and poorly written. Almost the entire text was redone in order to clarify the points raised in the revised manuscript, all changes are punctuated; 

Reviewer #3

Thank you for all the important recommendations, the introduction has been redone as well as the discussion has been rearranged to give more attention to the points raised. The grammar and spelling check was also done and to make the text clearer and more objective.

Reviewer #4

Thanks for all the important recommendations, the title has been modified, the data errors presented have been adjusted, as well as the introduction has been shortened, the methodology has been adjusted in order to meet all the recommendations raised. The contributions were very useful and valuable.

---

## [Decision Letter · Decision Letter 1]

24 May 2021

PONE-D-20-32227R1

Hypertension induced by pregnancy and neonatal outcome: Results from a retrospective cohort study in preterm under 34 weeks.

PLOS ONE

Dear Dr. ROCHA,

Thank you for submitting your manuscript to PLOS ONE. After careful consideration, we feel that it has merit but does not fully meet PLOS ONE’s publication criteria as it currently stands. Therefore, we invite you to submit a revised version of the manuscript that addresses the points raised during the review process.

We look forward to receiving your revised manuscript.

Kind regards,

Antonio Simone Laganà, M.D., Ph.D.

Academic Editor

PLOS ONE

Journal Requirements:

Additional Editor Comments (if provided):

Authors have performed the required changes, improving significantly the quality of the article.

Nevertheless, one of the Reviewers still has some minor concerns: for this reason, I invite the authors to perform these minor revision.

Reviewers' comments:

Reviewer's Responses to Questions

**Comments to the Author**

1. If the authors have adequately addressed your comments raised in a previous round of review and you feel that this manuscript is now acceptable for publication, you may indicate that here to bypass the “Comments to the Author” section, enter your conflict of interest statement in the “Confidential to Editor” section, and submit your "Accept" recommendation.

Reviewer #1: (No Response)

2. Is the manuscript technically sound, and do the data support the conclusions?

Reviewer #1: Yes

3. Has the statistical analysis been performed appropriately and rigorously? 

Reviewer #1: Yes

4. Have the authors made all data underlying the findings in their manuscript fully available?

Reviewer #1: Yes

5. Is the manuscript presented in an intelligible fashion and written in standard English?

Reviewer #1: Yes

6. Review Comments to the Author

Reviewer #1: Thanks for your efforts to improve this paper, and it is much better now. I still had a few minor comments on the attached PDF. One concern was that your occasionally refer to hypertensive disorder as "present". Though you define pre-eclampsia, HELLP, etc., you do not define "present". I request that you do.

At least in USA, to "terminate" a pregnancy commonly means to Abort. I'd recommend alternative wording, e.g,. electively deliver, deliver prematurely, induce labor, etc.

Your data, especially reqarding PVL, suggests that more than 59.7% of the G1 moms might have benefited from Mag Sulfate; can you comment on that?

Also, contrary to ACOG recommendations, the Cesarean Section rate seems very high. Was there truly a "need", or was this a misperception among the OBs?

7. PLOS authors have the option to publish the peer review history of their article (what does this mean?). If published, this will include your full peer review and any attached files.

Reviewer #1: No

---

## [Author Response · Author response to Decision Letter 1]

17 Jul 2021

I am pleased to resubmit for publication the revised version of: “Hypertension induced by pregnancy and neonatal outcome: Results from a retrospective cohort study in preterm under 34 weeks.” , PLOS ONE-D-20-32227R1, under the authorship of , Marta David Rocha de Moura, Paulo Roberto Margotto, Karina Nascimento and Maria Rita CG Novaes for your appreciation. All suggested suggestions were evaluated and considered. In this context, the manuscript, Response to Reviewers and Revised Manuscript witTrack Changes were attached.

---

## [Editor Report · Decision Letter 2]

26 Jul 2021

Hypertension induced by pregnancy and neonatal outcome: Results from a retrospective cohort study in preterm under 34 weeks.

PONE-D-20-32227R2

Dear Dr. ROCHA,

We’re pleased to inform you that your manuscript has been judged scientifically suitable for publication and will be formally accepted for publication once it meets all outstanding technical requirements.

Kind regards,

Antonio Simone Laganà, M.D., Ph.D.

Academic Editor

PLOS ONE

Additional Editor Comments (optional):

I carefully evaluated the revised version of this manuscript.

Authors have performed the required changes, improving significantly the quality of the paper.
---

## [Editor Report · Acceptance letter]

5 Aug 2021

PONE-D-20-32227R2 

Hypertension induced by pregnancy and neonatal outcome: Results from a retrospective cohort study in preterm under 34 weeks. 

Dear Dr. Rocha de Moura:

I'm pleased to inform you that your manuscript has been deemed suitable for publication in PLOS ONE. Congratulations! Your manuscript is now with our production department. 

Kind regards, 

on behalf of

Dr. Antonio Simone Laganà 

Academic Editor

PLOS ONE